# Methionine uptake via the SLC43A2 transporter is essential for regulatory T-cell survival

Neetu Saini*, Afsana Naaz*, Shree Padma Metur*, Pinki Gahlot*, Adhish Walvekar, Anupam Dutta, Umamaheswari Davathamizhan, Apurva Sarin, Sunil Laxman

**Cell death, survival, or growth decisions in T-cell subsets depend on interplay between cytokine-dependent and metabolic processes. The metabolic requirements of T-regulatory cells (Tregs) for their survival and how these are satisfied remain unclear. Herein, we identified a necessary requirement of methionine uptake and usage for Tregs survival upon IL-2 deprivation. Activated Tregs have high methionine uptake and usage to S-adenosyl methionine, and this uptake is essential for Tregs survival in conditions of IL-2 deprivation. We identify a solute carrier protein SLC43A2 transporter, regulated in a Notch1-dependent manner that is necessary for this methionine uptake and Tregs viability. Collectively, we uncover a specifically regulated mechanism of methionine import in Tregs that is required for cells to adapt to cytokine withdrawal. We highlight the need for methionine availability and metabolism in contextually regulating cell death in this immunosuppressive population of T cells.**

## Introduction

T cells play central roles in adaptive immune responses. To mount appropriate immune responses, T-cell subsets must adapt to a range of extracellular nutrient levels and environmental cues. Several signals push T cells out of quiescence and toward acquiring new functions. In this context, it is clear that metabolic reprogramming is central to the survival, differentiation, or functions of T cells and works in concert with key signaling systems and cytokines (Yang et al, 2013; Buck et al, 2015; He et al, 2017; Bantug et al, 2018; Chapman et al, 2020). What the different metabolic requirements are in distinct T-cell subtypes and how these are satisfied remain unclear. Depending upon the T-cell subset, distinct nutrients such as glucose, amino acids, or lipids control metabolic outputs that influence immune signaling and regulates the function and proliferation of T cells (Buck et al, 2015; Walls et al, 2016; Chapman et al, 2020). In general, T-cell activation in response to an antigen results

in transcriptional and metabolic remodeling, leading to new functions which include the production of cytokines and molecules that support T-cell survival, expansion, and functional demands (Fox et al, 2005; MacIver et al, 2013; Blagih et al, 2015; Buck et al, 2015; Menk et al, 2018; Cho et al, 2019). Because T-cell subsets function in diverse and dynamic niches, they require different energetic and metabolic pathways for their survival and function (O'Neill et al, 2016; Scharping et al, 2016; Pan et al, 2017; Siska et al, 2017). For example, upon antigen activation, CD4+ T-effector cells switch from oxidative phosphorylation (OXPHOS) to glycolysis to meet metabolic demands, whereas antigen-activated CD4+CD25hiFoxp3+ T-regulatory cells (Tregs) switch to lipid oxidation and low glycolysis (Wang et al, 2011; He et al, 2017; Siska et al, 2017). Such differing uses of biosynthetic pathways distinguish T-cell fate choices. In this context, what the metabolic requirements that control Tregs survival are and how the cell satisfies them remain an important yet poorly explored area.

Within this framework, multiple amino acids have unique roles in controlling T-cell function. Amino acids have diverse roles in metabolism and signaling and control multiple cellular programs. Studies in naïve T cells noted substantial changes in amino acid pools compared with activated T cells (Ananieva et al, 2014; Marchingo & Cantrell, 2022). Several amino acids such as leucine, glutamine, arginine, and tryptophan can regulate T-cell homeostasis and function, and the expression of distinct amino acid transporters is critical for these functions (Baban et al, 2009; Cobbold et al, 2009; Yan et al, 2010; Geiger et al, 2016; Ma et al, 2017). In T effectors (Teffs), critical roles for neutral and branched-chain amino acids were identified during T-cell expansion, for mTORC1 activation and for meeting requirements for bioenergetics (Sinclair et al, 2013, 2019; Ma et al, 2017). In Teffs, the solute carrier transporter SLC7A5 (which transports large neutral amino acids) was specifically up-regulated during activation, and this was required for their expansion in amino acid dependent contexts (Hayashi et al, 2013; Sinclair et al, 2013). CD4+ and CD8+ T cells deficient in SLC7A5 showed impaired clonal expansion and effector functions (Hayashi et al, 2013; Sinclair et al, 2013) and reduced mTORC1 activation concurrent with decreased glutamine and glucose uptake after T-cell activation (Sinclair et al, 2013). More recently, multifaceted roles for methionine were identified for

Institute for Stem Cell Science and Regenerative Medicine (DBT-inStem), Bengaluru, India

Correspondence: sarina@instem.res.in; sunil@instem.res.in
*Neetu Saini, Afsana Naaz, Shree Padma Metur, and Pinki Gahlot contributed equally to this work.

T-cell activation and differentiation/expansion into Teffs (Sinclair et al, 2019). In the Teffs context, methionine-transport regulated by the SLC7A5 transporter was the rate-limiting, essential factor in the proliferation and differentiation of these T cells (Sinclair et al, 2019). Similarly, branched-chain amino acids transported by SLC3A2 and CD98 regulate activation and suppressor function in Tregs (Ikeda et al, 2017; Shi et al, 2019). These data suggest the likely existence of many as yet unknown roles of amino acids and amino acid transporters in regulating T-cell fates.

Tregs are crucial for peripheral tolerance and immune homeostasis (Vignali et al, 2008; Corthay, 2009). Their immunosuppressive function is critical for pathologies such as autoimmunity, cancer, and tissue damage (Josefowicz et al, 2012; Ohkura et al, 2013). Tregs have unique metabolic requirements that are distinct from other T cells like the Teffs (Michalek et al, 2011; He et al, 2017; Yang et al, 2017). Tregs survive and protect themselves in dynamic and nutrient-limiting microenvironments, and these survival decisions have been best studied in the context of cytokine requirements in these cells. Therefore, understanding how Tregs regulate survival/death programs becomes critical to more completely address how they function. One mechanism by which Tregs regulates effector CD4[+] T-cell numbers is by depleting cytokines IL-2 in the microenvironment (Pandiyan et al, 2007; Höfer et al, 2012). Relevantly, activated CD4[+] and CD8[+] T effectors require cytokines like IL-2 for their survival and undergo apoptosis upon its withdrawal (Boyman et al, 2007; Purushothaman & Sarin, 2009). However, Tregs continue to survive in cultures without IL-2 for extended periods of time (Perumalsamy et al, 2012), and this indicates that additional, as yet unknown, factors enable the survival of Tregs. In this context, noncanonical (cytoplasmic) Notch1 activity can control Tregs survival, by regulating mitochondrial activity, and metabolism (Perumalsamy et al, 2012; Saini et al, 2022). Unlike in Teffs where Notch1 is localized in the nucleus, in Tregs Notch1 (which carries out protective effects) is predominantly cytoplasmic (Perumalsamy et al, 2012). Although these roles of Notch1 in enabling Tregs survival are critical, the unique metabolic requirements of Tregs for their survival and how these metabolic requirements are satisfied remain unknown.

Herein, we report a unique amino acid requirement for Tregs survival when IL-2 is deprived. Upon IL-2 withdrawal, Tregs cells exhibit an increased requirement of a specific amino acid, methionine, which is critical for cell survival. This methionine uptake is enabled by the Notch1-mediated regulation of the activity of a specific solute carrier (SLC) transporter in Tregs. The transporter-mediated uptake and subsequent use of methionine is the limiting factor for Tregs survival. We thus identify an essential amino acid requirement for Tregs survival upon IL-2 withdrawal. This process is mediated by a novel Notch1-dependent amino acid transporter axis that regulates the sustained supply of methionine and thereby Tregs survival.

# Results

## Activated Tregs uptake and metabolize methionine upon cytokine withdrawal

To investigate the status of amino acids in activated Tregs (referred to as Tregs in the text) survival contexts, we estimated changes in amino acid pools in murine Tregs upon IL-2 withdrawal. Relative steady-state concentrations of intracellular amino acids were quantitatively assessed over the first ~6 h after IL-2 withdrawal in Tregs, where they remained in complete medium with dialyzed serum (CMDS). Total metabolites were extracted from Tregs cultured in CMDS without IL-2 for 1, 3, and 6 h (Fig 1A: inset schematic), and amino acids were quantitatively assessed using targeted liquid chromatography/mass spectrometry (LC/MS/MS). Most of the amino acids did not show significant changes in their relative amounts (Fig 1A), indicating that Tregs continued to maintain an overall stable homeostasis with the external environment in this condition. Contrastingly, intracellular methionine amounts showed the most substantial decrease of all amino acids, decreasing as rapidly as within 1 h of cytokine withdrawal (Fig 1A). In addition, a decrease in the related sulfur amino acid cysteine, as well as histidine and tryptophan was also observed (Fig 1A). This suggested a possible shift in homeostasis in Tregs after IL-2 withdrawal, particularly toward increased methionine consumption. We therefore decided to subsequently focus on addressing a possible requirement specifically for methionine in Tregs post IL-2 withdrawal, based on the observation that methionine showed the most substantial decrease in amounts. In addition, we were intrigued by observations coming from a distinct context wherein methionine regulated CD4[+] T cells activation, differentiation, and proliferation (Geltink & Pearce, 2019; Sinclair et al, 2019) and wondered how this context was distinct.

Methionine is converted into its major metabolite SAM, and methyltransferases transfer the methyl group from SAM to produce a methylated substrate and SAH. We therefore next assessed the relative steady-state amounts of SAM and SAH, as part of the methionine cycle (Fig 1B). As observed with methionine levels shown earlier, from the same experiment, we observed that relative SAM amounts also significantly reduced in Tregs when cultured without IL-2 in complete medium (Fig 1B). As shown earlier, cysteine levels also showed a reduction (Fig 1B), consistent with expectations because cysteine and methionine are closely coupled to each other in the sulfur amino acid cycle. Together, these data suggest the possibility of an acute requirement of methionine in Tregs, upon IL-2 withdrawal, which is sustained by continuous uptake of this amino acid.

The previous data provides a "bulk estimate" of steady-state levels of these metabolites. The experiment cannot unambiguously address if there is high uptake of extracellular methionine and/or its increased consumption in these Tregs post IL-2 withdrawal. To definitively address these possibilities, a separate "flux" experiment using a pulsing of stable-isotope $^{13}C_5^{15}N$-labeled methionine was performed, using the experimental paradigm shown in Fig 1C. Activated Tregs were cultured in sulfur amino acid (SAA) dropout medium (DOSAA) (without IL-2 and with unlabeled methionine) for 1, 3, and 6 h. Twenty minutes before each of these time points, cells were washed with DOSAA, and to these cells, labeled methionine ($^{13}C_5^{15}N$) in DOSAA was added. After 20 min of labeled methionine addition, we collected cells and assessed the fraction of labeled versus unlabeled methionine in these cells (Fig 1D). Notably, we observed nearly complete replacement of unlabeled methionine with labeled methionine, with no detectable signal coming from the unlabeled methionine (which represents existing methionine pool

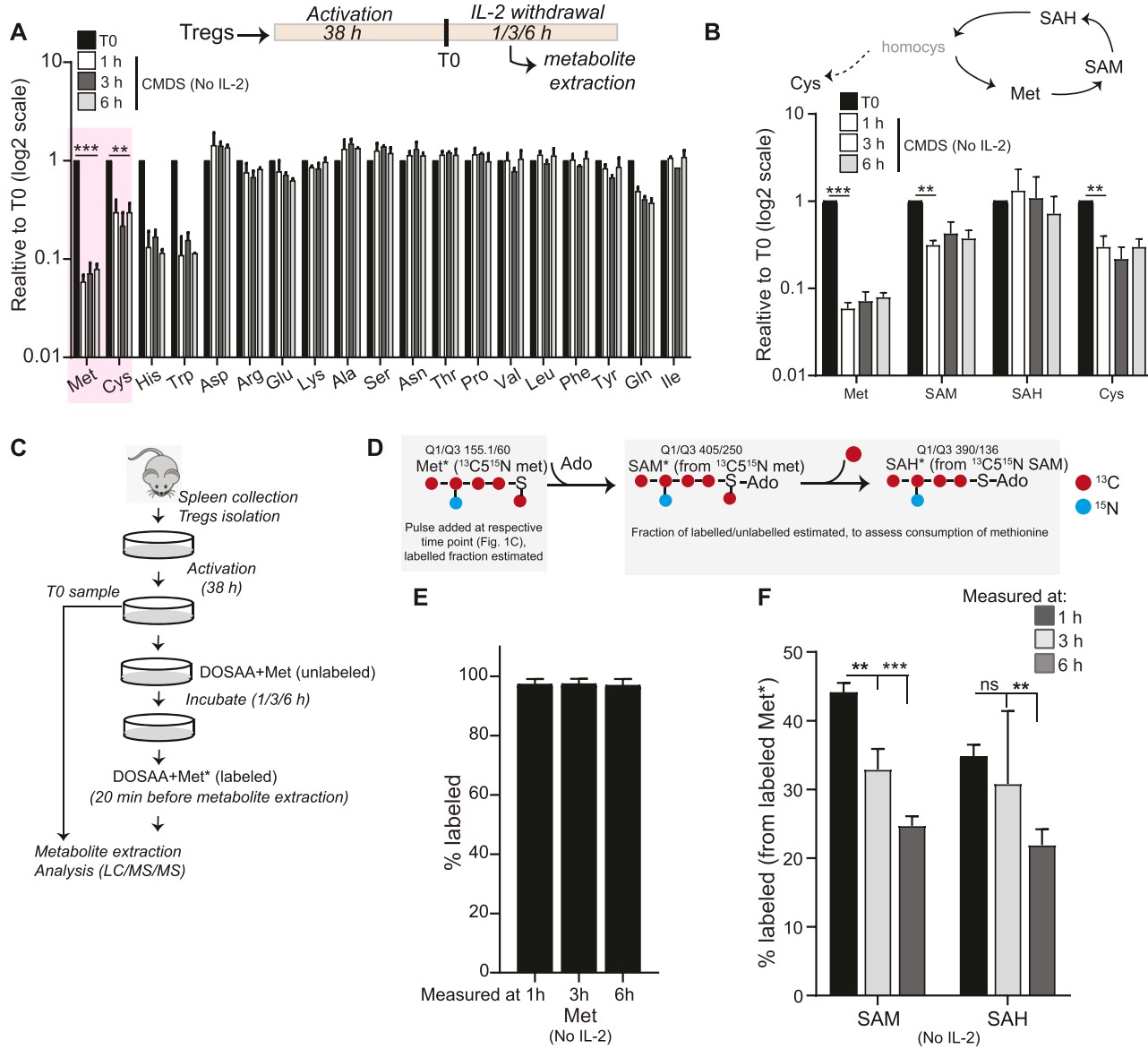

**Figure 1. Tregs have a specific requirement to take up and use methionine upon IL-2 withdrawal.**

**(A)** Amino acid changes in Tregs over time: Quantitative liquid chromatography–mass spectrometric (LC/MS/MS) analysis to determine relative changes in intracellular amino acid amounts in Tregs cultured in complete medium with dialyzed serum (CMDS) and without IL-2 for 1, 3, and 6 h. Amino acid amounts are plotted relative to T0 (onset of IL-2 withdrawal). Inset schematic: experimental timeline for assessment of metabolites in activated Tregs. **(B)** Methionine and sulfur cycle metabolites in Tregs: LC/MS/MS based analysis to determine relative changes in intracellular levels of methionine, SAM, SAH, and cysteine in Tregs cultured in CMDS without IL-2 for 1, 3, and 6 h. Data are plotted relative to T0 (also see Fig 1A for methionine and cysteine data). **(C)** Experimental approach to assess methionine uptake, and consumption in activated Tregs cultured in dropout of sulfur amino acids (DOSAA) and without IL-2. Tregs (in DOSAA+ unlabeled met) remained in culture until the specified time point. 20 min before the specified time point (1, 3, or 6 h) cells were washed in DOSAA, and then DOSAA+$^{13}$C5$^{15}$N–labeled methionine (150 $\mu$M) was added for 20 min, and cells were collected and processed for labeled metabolite analysis. **(D)** Schematic illustrating a stable-isotope label–based flux experiment for methionine uptake and methionine usage to SAM and SAH, at different time points after IL-2 withdrawal. $^{13}$C5$^{15}$N methionine in DOSAA was pulsed for 20 min at the indicated time point (see Fig 1C). After extraction, the fraction of methionine with and without the $^{13}$C5$^{15}$N label was quantitatively estimated by targeted, quantitative LC/MS/MS, to indicate the extent of uptake of methionine. Similarly, the extent of label incorporation coming from methionine to SAM and SAH was assessed, to estimate the amount of methionine converted to SAM and SAM to SAH at that indicated time point. **(E)** Stable-isotope–based assessment of methionine uptake: after adding $^{13}$C5$^{15}$N labeled methionine at the indicated time point, the fraction of labeled to unlabeled methionine was estimated. Note: at all time points, ~100% of the methionine estimated was $^{13}$C5$^{15}$N labeled, indicating that all stores of (unlabeled) methionine was consumed and replaced within this time. Also see Fig S1. **(F)** Stable-isotope–based assessment of methionine usage to SAM and SAH: for each time point indicated, after a 20-min addition of $^{13}$C5$^{15}$N-labeled methionine to Tregs, the amount of label incorporated into SAM and SAH was estimated and is indicated as the fraction labeled. A higher fraction labeled will indicate a greater consumption of methionine to SAM or SAH, in that period of time. **(A, B, E, F)** Data indicate mean ± SD of two independent experiments in panel (A) and (B) and three independent experiments in panel (E) and (F). ***$P$ ≤ 0.001 and **$P$ ≤ 0.01, ns implies nonsignificant. Also see Fig S1.

Source data are available for this figure.

in the cell) (Figs 1E and S1). These data indicate that Tregs continuously take up external methionine and use it rapidly. Next, to unambiguously assess relative rates of methionine consumption and usage, we quantified the amount of labeled methionine that is converted into labeled SAM and subsequently labeled SAH, in the same experimental set up. Notably, labeled SAM amounts (to unlabeled SAM) were highest in 1 h after IL-2 withdrawal and steadily decreased over 3 and 6 h (Fig 1F). Correspondingly, a similar trend was observed for SAH. These data indicate that in Tregs post IL-2 withdrawal, there is a high consumption of methionine to SAM (and SAM to SAH) and that the highest conversion of methionine to SAM occurs within this ~1 h period after IL-2 withdrawal.

Collectively, these data demonstrate that Tregs (post IL-2 withdrawal) continuously take up extracellular methionine and consume this methionine to synthesize SAM, which itself is used (producing SAH). The highest methionine consumption occurs over the first ~1 h after IL-2 withdrawal.

### Methionine is essential for the survival of Tregs in the absence of IL-2

Because activated Tregs survive after IL-2 deprivation (Perumalsamy et al, 2012), to investigate if methionine was required for this survival, Tregs were cultured without IL-2 either in SAA dropout or complete media with dialyzed serum for 24 h, and the extent of apoptotic damage was quantified. This was by using well established approaches to assess apoptosis in this exact experimental system, as shown earlier (Perumalsamy et al, 2009; Marcel & Sarin, 2016). Images indicating apoptotic or non-apoptotic nuclei from Tregs in these conditions are shown in Fig S2. The dropout of SAAs abrogated Tregs survival in the absence of IL-2 (Fig 2A). However, adding only methionine at the onset of IL-2 withdrawal (T0) rescued cells from apoptosis triggered by IL-2 withdrawal. In contrast, unlike methionine, the addition of cysteine to Tregs cultured in the absence of SAAs and IL-2 did not confer protection from apoptosis (Fig 2A). Furthermore, cysteine in combination with methionine provided the same protection to Tregs as methionine alone (Fig 2A). Together, these data suggest that the protective role of SAAs in Tregs is specific to methionine. To further assess this methionine requirement for Tregs survival, in complementary experiments, we supplemented cells with ethionine, an antimetabolite and antagonist of methionine. Ethionine competitively blocks methionine uptake and use. The addition of ethionine reduced the methionine-dependent cell survival after IL-2 withdrawal (Fig 2B). Finally, we better assessed the critical time window of methionine requirements for Tregs survival after IL-2 withdrawal. For this, we added back methionine at different time intervals (T0, T5, and T9) after IL-2 withdrawal and shifting cells to DOSAA. The protection of Tregs from apoptosis was restored to same extent as in Tregs cultured in CMDS where methionine was added back at the initial hours (i.e., T0 and T5) as compared with ~9 h after IL-2 withdrawal (T9) and shift to DOSAA (Fig 2C). As controls, we assessed the identity of Tregs generated from WT mice cultured with or without IL-2 either in SAA dropout or complete media, by immune staining for Foxp3 protein and found that Tregs in all these conditions retained Foxp3 expression (Fig 2D). Collectively, these data reveal a specific

requirement of methionine uptake and usage for Tregs survival post IL-2 withdrawal.

### Notch1 function is required for methionine uptake and survival of Tregs upon cytokine withdrawal

In IL-2–limiting conditions, prior studies had established that the activity of nonnuclear Notch1 enables the survival of Tregs (Perumalsamy et al, 2012). We therefore asked if the methionine-dependent survival of Tregs after IL-2 withdrawal also requires Notch1 activity. GSI-X (γ-secretase inhibitor-X), a pharmacological inhibitor of the enzyme γ-secretase was used to inhibit the cleavage and release of the Notch1 intracellular domain, as established in earlier studies (Perumalsamy et al, 2012; Saini et al, 2022). As a control, and consistent with earlier studies (Saini et al, 2022), Tregs cultured with GSI-X for 7 h showed a reduction in processed Notch1 protein and the canonical Notch1 transcriptional target Hes1 protein (Fig S3). Notably, the inhibition of Notch1 activity by GSI-X significantly reduced the protection conferred by methionine on Tregs survival (Fig 3A). It may be noted that to reduce any toxicity of GSI-X on Tregs, in these experiments, GSI-X and methionine were added 5 h after IL-2 withdrawal. Also note that in this experiment, the addition of methionine was not at the onset of IL-2 withdrawal as in earlier experiments. Therefore, just to ensure methionine-dependent protection of Tregs, we added a slightly higher concentration (200 µM) of methionine.

We next asked if the methionine-dependent protection of Tregs apoptosis depended on Notch1. For this, we used Tregs isolated from Notch1$^{+/+}$ (Cre−ve) or Notch1$^{-/-}$ (Cre+ve; Cd4-Cre::Notch1$^{lox/lox}$) mice and activated in vitro. Notch1$^{-/-}$ Tregs (unlike their Notch1$^{+/+}$ counterparts), undergo apoptosis when cultured without IL-2 for 24 h (Fig 3B). In this context, the addition of methionine failed to rescue the survival of Notch1-null Tregs (Fig 3B). Contrastingly, and consistent with Fig 2A–C, the addition of methionine at the onset of IL-2 withdrawal protected activated Notch1$^{+/+}$ (Cre−ve) Tregs from apoptosis when cultured in the absence of SAAs (DOSAA) and IL-2 (Fig 3B). Notch1$^{+/+}$ (Cre−ve) Tregs survived well when cultured in CMDS without IL-2 as compared with Notch1$^{-/-}$ (Cre+ve). Together, the data from these experiments show that Notch1 function is required for the methionine-mediated Tregs survival post IL-2 withdrawal. This also reaffirms earlier results (Perumalsamy et al, 2012) that demonstrate a requirement for Notch1 in Treg survival.

To further understand if Notch1 had a role in mediating methionine uptake in Tregs, we assessed changes in the mRNA levels of several amino acid transporters after IL-2 withdrawal, both in the presence and absence of GSI-X (3 h treatment). These transporters belong to the solute carrier (SLC) superfamily, which are major transporters of several amino acids including methionine (Pizzagalli et al, 2021). Among the transporter transcripts analyzed, SLC6A17, SLC1A5, and SLC7A8 showed very high Ct values (>30) (Fig 3C), indicating very low/basal transcript levels in Tregs. In contrast, SLC3A2, SLC43A1, SLC43A2, and SLC7A5 all had higher mRNA expression as assessed by Ct values (Fig 3C). Interestingly, SLC43A2 and SLC43A1 transcript levels increased in Tregs cultured without IL-2 in complete medium for 3 h (Fig 3D). Notably, abrogating Notch1 activity by using GSI-X in Tregs cultured without IL-2 decreased the mRNA levels of only the SLC43A transporter (Fig 3D). None of the other SLC transporters revealed any trend of

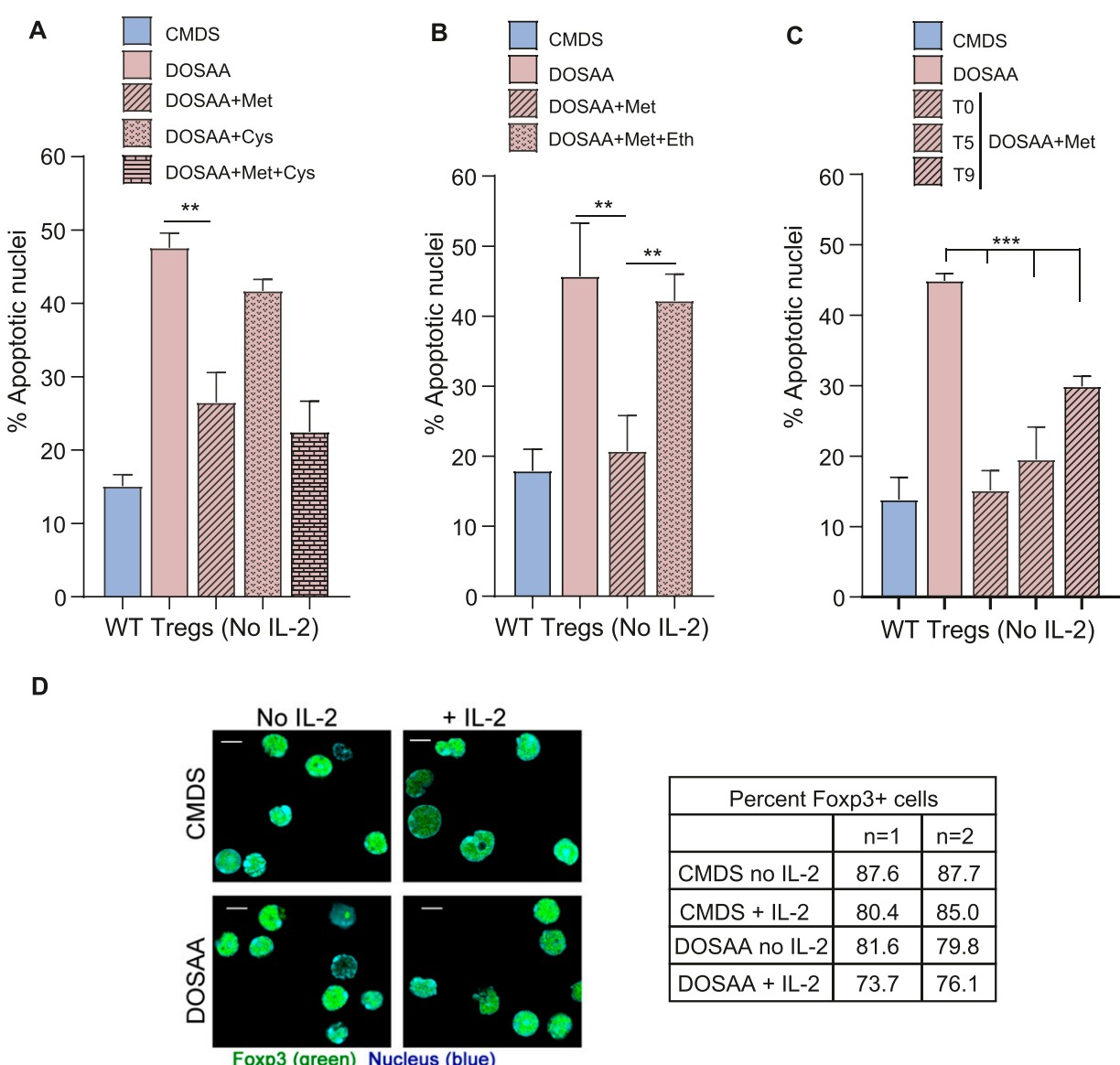

**Figure 2. Activated Tregs require methionine for survival upon IL-2 withdrawal.**
**(A)** Tregs survival and methionine: Percentage of cells with apoptotic nuclei in WT Tregs cultured without IL-2 for 18–22 h in complete medium with dialyzed serum (CMDS) and dropout of sulfur amino acids (DOSAA), without or with methionine (150 $\mu M$) or cysteine (150 $\mu M$) added as indicated at T0 (the onset of IL-2 withdrawal). Methionine and cysteine were used at a concentration of 100 $\mu M$ in combination. Also see Fig S2 which contains images of nuclear integrity in Tregs either undergoing or not undergoing apoptosis. **(B)** Effect of competitive inhibition of methionine uptake: indicated are the percentage of cells with apoptotic nuclei in WT Tregs cultured without IL-2 for 18–22 h in CMDS and DOSAA without and with methionine (150 $\mu M$) added at T0 or methionine in combination with the methionine antimetabolite ethionine (750 $\mu M$). Ethionine was added at T0 followed by addition of methionine after 20 min. **(C)** Methionine-supplementation based rescue of cell death: Indicated are the percentage of cells with apoptotic nuclei in WT Tregs cultured without IL-2 for 18–22 h in CMDS and DOSAA, with methionine (150 $\mu M$) added at T0 or after 5 h (T5) and 9 h (T9) of the onset of IL-2 withdrawal. **(D)** Expression of Foxp3 in WT Tregs, with the dropout of IL-2 and/or methionine: representative confocal images (at least 100 cells/condition) of activated WT Tregs immune-stained for Foxp3 (green) and nuclei (Hoechst 33342—blue). WT Tregs were cultured in CMDS and DOSAA with (1 $\mu g/ml$) or without IL-2 for 6 h. The table indicates the percentage of Foxp3+ WT Tregs for these conditions. Data represent two independent experiments. Scale bar 5 $\mu m$. Data indicate mean ± SD of three independent experiments. ***$P \leq 0.001$ and **$P \leq 0.01$.
Source data are available for this figure.

putative Notch1-dependent expression. In particular, the mRNA levels of SLC7A5, a transporter required for the import of methionine in CD4⁺ T cells (Sinclair et al, 2019), remained unaltered in Tregs. These data collectively suggested that the SLC43A transporters might have a role in the Notch1-mediated regulation of methionine uptake in Tregs.

## SLC43A2 transporter is essential for methionine-dependent Tregs survival

Of the transporter transcripts assessed, only the SLC43A family transporters had high transcript amounts, and SLC43A2 transcripts also showed a Notch1-dependent expression post IL-2 withdrawal.

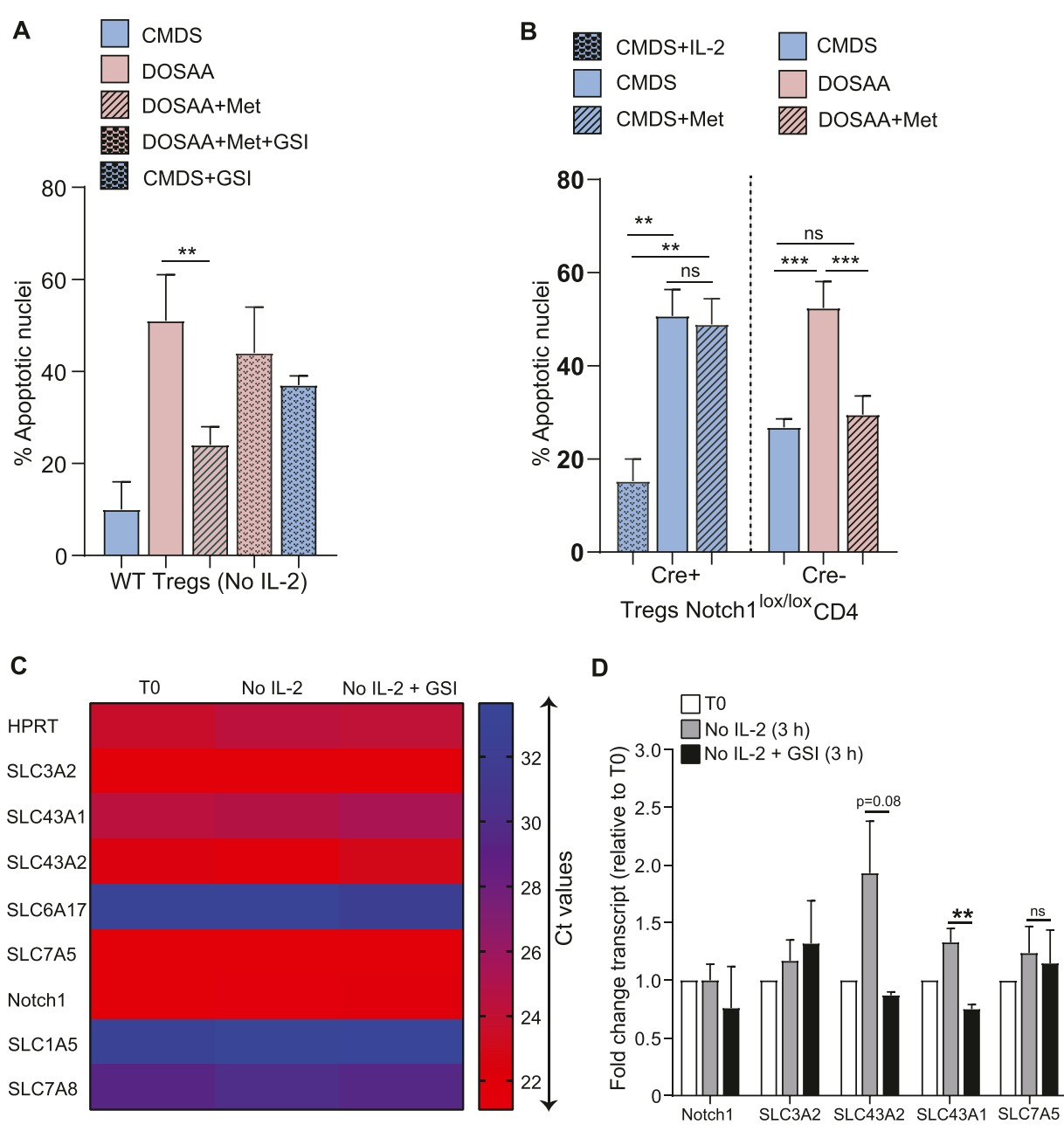

**Figure 3. Notch1 regulates methionine-dependent Tregs survival.**
**(A)** Notch1 inhibition abrogates methionine-dependent Tregs survival: assessment of apoptotic nuclei in Tregs in the presence (7.5 µM) or absence of GSI-X, used to inhibit Notch1. Cells were cultured without IL-2 in CMDS or dropout of sulfur amino acids (DOSAA). After 5 h, methionine (200 µM) and/or GSI were added to the indicated bars. Apoptosis was quantified after 18–22 h of IL-2 withdrawal. Also see Fig S3 (indicating GSI efficacy for Notch1 inhibition). **(B)** Notch1 requirement for methionine protection: (left of dashed division line) apoptotic nuclei in Notch1$^{-/-}$ (Cre+ve; Cd4-Cre::Notch1$^{lox/lox}$) Tregs when cultured for 18–22 h in CMDS with IL-2 (1 µg/ml), CMDS, or CMDS with methionine (150 µM). (Right of dashed division line) Apoptotic nuclei in Notch1$^{+/+}$ (Cre−ve) Tregs when cultured in the indicated conditions. (Note: Notch1$^{-/-}$ cells are in CMDS with or without methionine, and Notch1$^{+/+}$ cells are in DOSAA with or without methionine.) **(C)** Amino acid transporter transcript amounts in Tregs, without IL-2 and Notch1 inhibition with GSI-X: A heat map showing Ct values from RT-qPCR based analysis of transcripts of indicated SLC amino acid transporters in Tregs, assessed at T0 and post 3 h culture without IL-2 in the presence (10 µM) or absence of GSI-X. High Ct values indicate low transcript levels. **(D)** Relative changes in mRNA levels of the indicated SLC amino acid transporters in Tregs cultured for 3 h without IL-2 in the presence (10 µM) or absence of GSI-X. Fold change was measured relative to T0. **(A, B, C, D)** Data indicate mean ± SD of three independent experiments for panels (A) and (B), two independent experiments for panels (C) and (D). ***$P ≤ 0.001$ and **$P ≤ 0.01$, ns implies nonsignificant.

We, therefore, prioritized assessing possible roles of SLC43A2 and sought to understand if this transporter is required for Tregs survival upon IL-2 withdrawal, in the context of methionine availability. As an important comparison, we also assessed if SLC7A5, which regulates the import of methionine in CD4$^+$ T cells (Sinclair et al, 2019), had any role to play in Tregs survival in the absence of IL-2. Using shRNA, we knocked down, respectively, SLC43A2 and SLC7A5 in Tregs (Fig 4A). The shRNAs were extensively

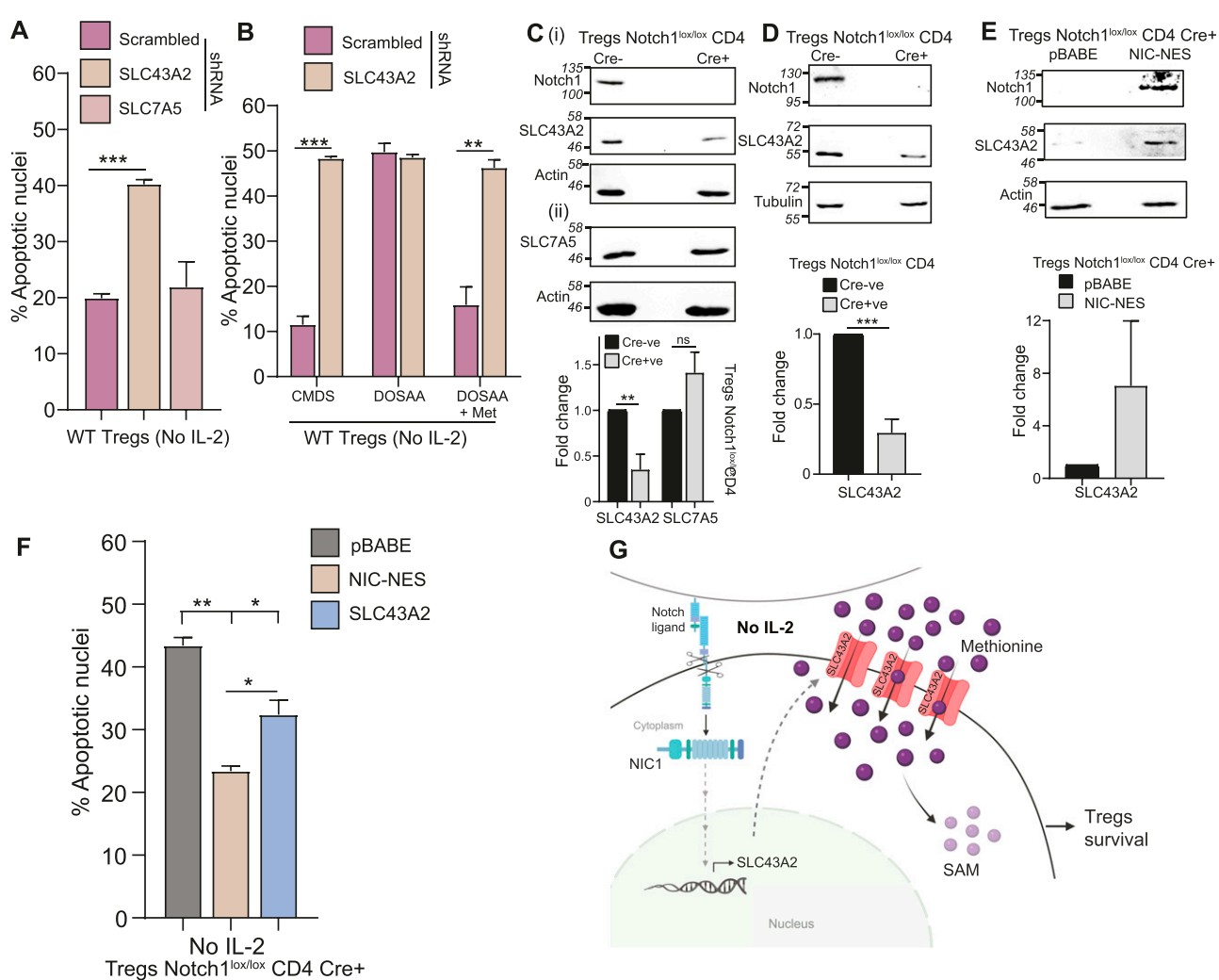

**Figure 4. Notch-dependent SLC43A2 transporter regulates methionine-dependent Tregs survival.**
**(A)** SLC43A2 and not SLC7A5 is required for Tregs survival: indicated are the percentage of cells with apoptotic nuclei in WT Tregs transduced with retroviruses containing shRNA to SCL43A2, SLC7A5, or scrambled control and cultured without IL-2 for 18–22 h. Also, see Fig S4A–C for shRNA-based knockdown efficacy and specificity. **(B)** SLC43A2 is required for methionine-dependent Treg survival: percentage of cells with apoptotic nuclei in WT Tregs transduced with retroviruses containing shRNA to SLC43A2 or scrambled control and cultured in complete medium with dialyzed serum (CMDS) or dropout of sulfur amino acids (DOSAA) with (150 $\mu$M) or without methionine for 18–22 h. Also, see Fig S4D for knockdown efficacy and Fig S4E for Foxp3 expression. **(C)** Notch1 dependence of SLC43A2 or SLC7A5 in activated Tregs: immunoblots of cell lysates prepared from activated Notch1$^{-/-}$ (Cre+ve; Cd4-Cre::Notch1$^{lox/lox}$) and Notch1$^{+/+}$ (Cre–ve) Tregs probed for (i) SLC43A2, Notch1, and (ii) SLC7A5 (actin control). The cell lysate was split and used for C-i and C-ii. Inset: densitometry based fold change in the protein levels of SLC43A2 and SLC7A5 in Notch1$^{-/-}$ Tregs relative to Notch1$^{+/+}$ Tregs. Data indicate mean ± SD of three independent experiments for C-i and two independent experiments for C-ii. Also, see Fig S5A and B showing steady-state SLC43A2 protein in Tregs and SLC43A2 or SLC7A5 protein in Teffs, in Notch1$^{-/-}$ versus Notch1$^{+/+}$. See Fig S5C for SLC43A2 and SLC7A5 protein in Tregs treated with the Notch1 inhibitor GSI-X. **(D)** SLC43A2 and SLC7A5 and Notch1 in freshly isolated Tregs: immunoblots probed for SLC43A2, Notch1 (tubulin loading control) in freshly isolated Notch1$^{+/+}$ and Notch1$^{-/-}$ Tregs. Inset: densitometric analysis indicating fold change in the protein levels of SLC43A2 in freshly isolated Notch1$^{+/+}$ and Notch1$^{-/-}$ Tregs. **(E)** Cytoplasmic Notch1 dependence: Immunoblots probed for SLC43A2, Notch1 (actin control) in Notch1$^{-/-}$ Tregs transduced with retroviruses containing pBABE-NIC-NES, or control pBABE plasmid (immunoblots are representative of three independent experiments). Inset: densitometry based fold change in the protein levels of SLC43A2 in Notch1$^{-/-}$ Tregs transduced with pBABE-NIC-NES relative to control pBABE. **(F)** Sufficiency of SLC43A2 for Tregs survival: indicated are the percentage of cells with apoptotic nuclei in Notch1$^{-/-}$ (Cre+ve; Cd4-Cre::Notch1$^{lox/lox}$) Tregs transduced with retroviruses containing pBABE-NIC-NES, pBABE-SLC43A2, or control pBABE plasmid and cultured without IL-2 for 18–22 h. Also, see Fig S5D for Notch1 and SLC43A2 protein levels in these conditions. **(G)** Model for methionine requirements for Tregs survival: Upon IL-2 depletion, Tregs continuously take up and use methionine for survival. This occurs through the SLC43A2 transporter in a Notch1-dependent manner. Notch1 regulates SLC43A2 expression, and this transporter allows Tregs to take up methionine, and use it, enabling Tregs survival. **(A, B, D, E, F)** Data indicate mean ± SD of two independent experiments in panels (B) and (F) and three independent experiments in panels (A), (D), and (E). ***$P \leq 0.001$, **$P \leq 0.01$, *$P \leq 0.05$, ns implies nonsignificant. Source data are available for this figure.

validated for efficacy and specificity (see the Materials and Methods section and Table S1). We further confirmed that the knockdown of SLC43A2 did not affect SLC7A5 transcript levels for nonspecific reasons (Fig S4A). Using shRNAs, SLC43A2, and SLC7A5 were ablated in Tregs (protein levels shown in Fig S4B and C), and Tregs survival

assessed after IL-2 withdrawal. The knockdown of SLC43A2, but not the knockdown of SLC7A5, specifically increased apoptotic damage as compared with Tregs transduced with scrambled control (Fig 4A). This suggests a necessary role for SLC43A2 in Tregs survival upon IL-2 withdrawal. We next asked whether the SLC43A2 transporter protein

was required for the methionine-dependent survival of Tregs in IL-2–limiting conditions. Tregs were transduced with scrambled (control) and SLC43A2 shRNA (Fig S4D). The knockdown of SLC43A2 abrogated Tregs survival when cultured without IL-2 in CMDS and in methionine supplemented medium (Fig 4B). These data are in contrast to wild-type cells where methionine supplementation increased cell survival. SLC43A2 abrogated Tregs retained Foxp3 expression, as assessed by immune staining (Fig S4Ei and ii), confirming that the knockdown of SLC43A2 does not alter Tregs identity as assessed based on this indicator. These data, therefore, show that the SLC43A2 transporter has a specific, necessary role in Tregs in enabling methionine-dependent survival in IL-2-limited medium.

Because Notch1 could not protect Tregs in the absence of methionine, and the SLC43A2 mRNA levels appeared to be Notch1-dependent (Fig 3D), we directly asked whether Notch1 regulates SLC43A2 protein levels in Tregs, and this thereby enables methionine-dependent survival. As important controls, we asked if SLC7A5, which is important for methionine import for the survival of CD4$^+$ cells, had any Notch1 dependency in Tregs. For this, the levels of the SLC43A2 and SLC7A5 proteins in activated Notch1$^{+/+}$ and Notch1$^{-/-}$ Tregs were determined. Notch1$^{-/-}$ Tregs showed a significant reduction in SLC43A2 protein levels relative to Notch1$^{+/+}$ Tregs (Fig 4Ci). In contrast, no change was observed in the SLC7A5 transporter protein levels (Fig 4Cii). Furthermore, we also compared SLC43A2 protein abundance in Notch1$^{-/-}$ Tregs as compared with Notch1$^{+/+}$ Tregs cultured without IL-2 for 4 h (Fig S5A). The SLC43A2 protein levels remain constitutively lower in the Notch1-null cells, consistent with a requirement for Notch1 to maintain SLC43A2 protein in Tregs. Notably, the protein levels of the SLC43A2 transporter or the SLC7A5 (which is important for methionine transport in CD4$^+$ effector T-cells) transporter did not change in activated Notch1$^{+/+}$ (Cre−ve) and Notch1$^{-/-}$ (Cre+ve) Teffs (Fig S5B). Consistent with these results, Notch1 inhibition by GSI also reduced SLC43A2 protein amounts but did not affect SLC7A5 protein levels in activated Tregs when compared with control (DMSO treated) cells (Fig S5Ci and ii). These data collectively indicated that the Notch1-mediated regulation of SLC43A2 is specific to Tregs. Finally, freshly isolated Notch1$^{-/-}$ Tregs showed decreased SLC43A2 protein levels, as compared with Notch1$^{+/+}$ Tregs (Fig 4D). This also further reiterates that the difference in the SLC43A2 protein level is not because of differential activation of Notch1$^{+/+}$ (Cre−ve) and Notch1$^{-/-}$ (Cre+ve) Tregs. Note: the loss of Notch1 (Notch1-null) does not alter the expression/frequency of Tregs, but these cells show defects in suppressor function (see Marcel and Sarin [2016] and Saini et al [2022]). Together, these data suggest that amount of SLC43A2 transporter protein depends on Notch1 availability, and the loss of Notch1 (and not IL-2) reduces SLC43A2.

As an additional control, we analyzed the SLC43A2 promoter for the putative RBP-jk–binding site consensus motif (CGTGGGAA) (Tun et al, 1994) in the promoter region, covering at least ~3,000 bp upstream of transcription initiation site. However, the SLC43A2 promoter region does not have any RBP-jk–binding site. This hinted at possible noncanonical or indirect Notch1 function regulating SLC43A2 amounts. Separately, in these exact conditions, earlier studies had established that the Notch1 intracellular domain (NIC) activity from the cytoplasm protects Tregs from apoptosis in IL-2–limiting conditions (Perumalsamy et al, 2012). This was

established by transducing Notch1$^{-/-}$ Tregs with recombinant NIC tagged to a nuclear export signal to enrich localization of NIC in the cytoplasm of Tregs (Shin et al, 2006; Perumalsamy et al, 2012). We, therefore, examined the effect of NIC-NES on SLC43A2 protein levels, to determine if similar noncanonical Notch1 functionality is required for the SLC43A2 function in this methionine uptake context in Tregs. The overexpression of NIC-NES by retroviral transduction in Tregs (where Notch1 is highly enriched in the cytoplasm [Perumalsamy et al, 2012]) specifically increased protein levels of SLC43A2 in Notch1$^{-/-}$ Tregs (Fig 4E). Finally, we asked if the overexpression of SLC43A2 in Notch1$^{-/-}$ Tregs was in itself sufficient to protect these cells from apoptosis triggered by IL-2 withdrawal and if Notch1 had additional roles in enabling Tregs survival. Overexpressing SLC43A2 only partially rescued Notch1$^{-/-}$ Tregs from apoptosis (Fig 4F). Note: for activated Notch1$^{-/-}$ Tregs transduced with SLC43A2 and NIC-NES and pBABE vector control, the overexpression of NIC-NES and SLC43A2 in Notch1$^{-/-}$ was verified (Fig S5D). The rescue by overexpressing SLC43A2 was not equivalent to that provided by overexpression of NIC-NES (Fig 4F), suggesting additional factors in the survival program that are under Notch1 regulation. Collectively, these data show that SLC43A2 is required for the methionine-dependent survival of Tregs in IL-2 deficient conditions, in a Notch1-dependent manner.

## Discussion

Understanding processes that enable death-survival decisions in T cells is a prerequisite to decipher how T cells regulate adaptive immune responses. Based on our data, we present a simple model for a necessary requirement for Tregs survival after IL-2 withdrawal (Fig 4G). Upon IL-2 withdrawal, Tregs require the sustained uptake, transport, and usage of methionine. To enable this, Notch1 (nonnuclear) functions to regulate the expression of a specific solute carrier transporter, SLC43A2. SLC43A2 allows Tregs to take up methionine, which is subsequently used (Fig 4G). A reduction in SLC43A2 protein because of reduced Notch1 activity in these contexts abolishes Tregs survival conferred by methionine. At this stage, these data suggest a coordinated "top-down" (Shyer et al, 2020) metabolic survival signaling and metabolic cascade in Tregs, where the IL-2 withdrawal and the activity of Notch1 together coordinate a methionine-dependent survival program.

To understand the full metabolic programs that regulate cell death/survival/growth programs, it is useful to separately question what metabolites are contextually needed and how these metabolic needs are sustained, versus what are the roles of specific metabolic programs in cells. In this study, we have only identified methionine uptake and usage as a limiting step in Tregs survival upon IL-2 withdrawal, with the SLC43A2 transporter sustaining this supply of sufficient methionine. This study, therefore, addresses a critical metabolic requirement needed for Tregs survival and how this requirement is satisfied. The SLC class of transporters constitutes a large superfamily with increasingly important metabolic roles (Schumann et al, 2020; Song et al, 2020), of which the more selective amino acid transporters (such as SLC7 or SLC43A) remain poorly studied. Currently, the critical roles of SLC class transporters for other metabolites, particularly glucose import via GLUT or SLC2A

transporters in controlling T-cell metabolism to enable differentiation or activation, are better known (Jacobs et al, 2008; MacIver et al, 2008; Basu et al, 2015; Gerriets et al, 2015). As gatekeepers of nutrient (including amino acid) availability and therefore enablers of metabolic programs, it is likely that the amino acid transporting SLC proteins will also play roles in the survival or development of variety of cells and likely also cross-talk with the glucose-dependent responses. Understanding how these transporters function in different contexts and how they are regulated by cytokine-dependent and nutrient signaling systems will be necessary to further understand how unique metabolic programs are enabled in T cells. Separately, it will be important to understand how T-cell fate–regulating signaling systems such as mTORC1, Akt, or Notch control the activity or functions of these transporters, to sustain specific metabolic programs that determine activation, differentiation, or survival/death decisions in lymphocytes. For example, in other cellular contexts, earlier studies have suggested that this form of NIC-mediated Notch1 function can converge on mTORC1 function to enable cell survival (Perumalsamy et al, 2012). In all likelihood, there will be contextual differences in the specific mechanisms of apoptosis protection mediated by methionine. All of these form exciting future directions of research.

Finally, the various possible functions of methionine in enabling Tregs survival remain unknown. Methionine, primarily through its metabolic product SAM, controls a variety of fundamental metabolic, signaling and regulatory processes in various cells (Walvekar et al, 2018b; Sanderson et al, 2019; Walvekar & Laxman, 2019; Lio & Huang, 2020; Roy et al, 2020). The presence or absence of methionine itself triggers cascading, hierarchically organized metabolic programs in cells (including glucose and other amino acid metabolism) (Walvekar et al, 2018b; Roy et al, 2020), to alter proliferation versus survival trajectories. In the context of T cells, in a relevant example of T effectors, antigenic stimulation–induced methionine uptake and usage appears to drive a program of nucleotides and proteins methylation for multiple (and not a single) fundamental processes (Sinclair et al, 2019). This is a contrasting context where cell growth and proliferation predominates (Sinclair et al, 2013; Geltink & Pearce, 2019), in contrast to the present study where for Tregs survival (where there is no growth or differentiation), cells require substantial methionine uptake and consumption, to presumably enable a distinct survival program. Reports in T helper ($T_h$) cells find that methionine metabolism maintains SAM pools to shape cell proliferation in part via altering histone modifications (Roy et al, 2020). It is therefore apparent that the contextual use of methionine in these different T-cell populations can have important yet poorly identified roles in regulating various functions. Therefore, understanding the methionine-dependent metabolic programs in Tregs and how they regulate signaling and regulatory outputs to regulate Tregs functions for adaptive immune responses will be an exciting direction of future enquiry.

# Materials and Methods

## Mice

C57BL/6J mice were procured from the Jackson Laboratory. The Notch1$^{lox/lox}$ and Cd4-Cre::Notch1$^{lox/lox}$ (Notch1$^{-/-}$) mouse strains were a kind gift from F Radtke (École Polytechnique Federale de Lausanne [EPFL]) (Wolfer et al, 2001). The Institutional Animal Ethics Committees of the National Centre for Biological Sciences (NCBS) and the Institute for Stem Cell Science and Regenerative Medicine (inStem), Bangalore, India, approved procedures involving mice, as per the guidelines of the Committee for the Purpose of Control and Supervision of Experiments on Animals, Government of India. Breeding colonies were maintained in-house (NCBS/inStem Animal Care and Resource Center) in controlled environments and in high-barrier specific pathogen-free conditions. Colonies were regularly tested for the global pathogen panel (based on FELASA recommendations).

## T-cell subsets isolation and activation

Tregs and Teffs were isolated from spleens of 8–12-wk-old mice of mixed gender as described previously (Saini et al, 2022). CD4$^+$CD25$^+$Tregs and CD4$^+$CD25$^-$Teffs were isolated using the Dynabeads FlowComp Mouse CD4$^+$CD25$^+$Treg cells Kit (11463D; Invitrogen) following the manufacturer's instructions and as previously described (Perumalsamy et al, 2012). Approximately 120–150 × 10$^6$ lymphocytes cells were used to isolate ~2 × 10$^6$ Tregs and ~12 × 10$^6$ Teffs per column. Tregs and Teffs were activated at a density of ~2 × 10$^6$/ml with magnetic beads (Invitrogen) coated with CD3 and CD28 antibodies (20 μl/ml) in a 24-well plate. Note: no IL-2 or TGF-β was added during the activation process. After 38–44 h, beads were removed by magnetic separation, and activated Tregs were washed and used for experiments. Cells were cultured in RPMI 1640 (Thermo Fisher Scientific) supplemented with 0.1% penicillin/streptomycin, glutamine, and 5% heat-inactivated fetal bovine serum (Complete Medium, CM) or 5% heat-inactivated fetal bovine dialyzed serum (CMDS) whenever required. When required, T cells were cultured in RPMI without L-cysteine, L-cystine, L-methionine (MP Biomedicals) supplemented with 0.1% penicillin/streptomycin, glutamine and 5% heat-inactivated fetal bovine dialyzed serum (DOSAA).

## Cell lines

The HEK293T (HEK) and NIH3T3 cell lines were obtained from the American Type Culture Collection (ATCC). Cell lines were cultured in DMEM (Thermo Fisher Scientific) supplemented with 0.1% penicillin/streptomycin, glutamine and 10% heat-inactivated fetal bovine serum (Scientific Hyclone TM). HEK cells were used for retroviral packaging till passage 20. Cells were maintained at 37°C with 5% CO$_2$. The cells were routinely tested free from mycoplasma by using the MycoAlert Kit (Lonza).

## Reagents and antibodies

### Reagents

TRIzol (15596026) and SYBR Green Master Mix (Thermo Fisher Scientific); PrimeScript first strand cDNA Synthesis Kit (6110A) (Takara Bio); Recombinant IL-2 (R&D Systems); magnetic beads coated with anti-CD3 and anti-CD28 antibodies (Invitrogen); γ-Secretase Inhibitor-X (GSI-X, 565771) and puromycin (508838) (Calbiochem-Merck Millipore); X-treme-GENE (6366236001), histopaque (10831), L-ethionine (E1260), L-cysteine (168149), and L-methionine (M9625) (Sigma-Aldrich); 13C515N-labeled L-methionine (CNLM-759-H-0.1) (Cambridge Isotope Laboratories, Inc.). All other reagents were

purchased from Himedia or Sigma-Aldrich. L-Ethionine, L-cysteine, and L-methionine stocks were made in autoclaved distilled water, filter sterilized, and stored at –80°C. 1 mM GSI-X was made in dimethylsulfoxide and stored at –20°C. Fresh aliquots were used for each experiment.

### Antibodies

PA5-71365 against SLC7A5 (Life Technologies/Thermo Fisher Scientific); 11-5773-80 against Foxp3 (eBiosciences/Thermo Fisher Scientific); ab186444 against SLC43A2 and mN1A (128076) against Notch1 (Abcam); AB15470 against Hes-1 (Millipore Sigma), ACTN05, MS-1295-P to actin and MS-581-P0 to tubulin (Neomarker); 11-4321-80 isotype control (eBiosciences/Thermo Fisher Scientific); horseradish peroxidase–linked anti-mouse (7076S) and anti-rabbit IgG (7074P2) (Cell Signaling Technology).

### Plasmids

Verified murine shRNA plasmids to SLC43A2 (TR5143399), SLC7A5 (TG513945), scrambled control shRNA (TR30012), and pVMV6 entry-SLC43A2 Human Tagged ORF Clone (RC206639) were from Origene. The set of four shRNA constructs supplied were screened individually for knockdown of protein expression in NIH3T3 (ATCC) cells. The construct[s] (usually one or two) that effectively knocked down the target protein was used in the retroviral transductions of T-cells. The target shRNA sequence used in the study is provided in Table S1.

The pBABE and pBABE-NIC-NES plasmids were gifts from BA Osborne (University of Massachusetts/Amherst). The Human SLC43A2 gene from pCMV6 entry vector was subcloned into the pBABE vector. Primers used for subcloning are listed in Table S1. Construct sequences were verified by automated Sanger sequencing conducted in-house.

### Retroviral transductions

HEKs ($0.25 \times 10^6$) were seeded in 35-mm dishes (Greiner Bio-one). Cells (50–60% confluent) were transfected the next day with retroviruses containing the gene of interest (1.5 $\mu$g) and packaging vector pCL-Eco (1.5 $\mu$g for plasmids and 2 $\mu$g for shRNA) using X-tremeGENE. Retrovirus transduction was as described earlier (Saini et al, 2022). Briefly, the virus in the supernatant was concentrated by centrifugation at 21,000$g$ for 1 h 30 min at 4°C. Tregs stimulated with anti-CD3−CD28–coated beads for 24 h were infected using a cocktail of the concentrated virus, RPMI-CM, 10 mM Hepes, and 8 $\mu$g/ml sequa-brene by centrifugation at 600$g$ for 90 min at 25°C. Twenty-four hours postinfection, Tregs were harvested, beads removed by magnetic separation, and continued in culture ($0.5 \times 10^6$ cells/ml) in RPMI-CM supplemented with IL-2 (1 $\mu$g/ml) to minimize stress to the cells. After 18–24 h, puromycin (1 $\mu$g/ml) was added to the media along with IL-2 (1 $\mu$g/ml) to enrich transfected cells. 48 h after the addition of puromycin, live cells were selected over histopaque (1.083 g/ml density) by density-gradient centrifugation at 300$g$ for 20 min. Cells (~$0.6$–$0.8 \times 10^6$) were washed twice in RPMI-CM and PBS and cultured in media supplemented with IL-2 (1 $\mu$g/ml) and IL-7 (2 ng/ml) for another 22–24 h and used for apoptosis assays. Knockdown or overexpression of targeted genes was assessed by Western blot analysis.

### Western Blot analysis

Cells ($0.4$–$0.6 \times 10^6$) were lysed in 25 $\mu$l SDS lysis buffer, and cell lysates were resolved by 10% SDS–polyacrylamide gel electrophoresis followed by Western blot analysis as described earlier (Saini et al, 2022). Primary antibodies were used at the following dilution: Notch1 (1:500), SLC43A2 (1:500), SLC7A5 (1:500), actin (1:1,000), HES1 (1:500), and tubulin (1:1,000) diluted in 5% skimmed milk in Tris-buffered saline−Tween 20 (TBST). Horseradish peroxidase–conjugated secondary antibody was used at a 1:1,000 dilution. The membranes were developed using Super Signal West Dura substrate (Thermo Fisher Scientific), and images were acquired in iBright FL1000 (Invitrogen).

### Metabolic profiling

Tregs ($2 \times 10^6$, activated for 38–40 h) were cultured in CMDS without IL-2 for 1, 3, and 6 h. After the incubation, the cells were pelleted down at 600$g$ for 3 min, and metabolites extracted. Briefly, 1 ml of ice-cold 10% methanol was added without disturbing the pellet (to quench metabolism) and further centrifuged at 600$g$ for 3 min at 4°C. Furthermore, 1 ml of 80% methanol (maintained at –45°C) was added to the pellet, vortexed for 15 s, and incubated at –45°C for 15 min for metabolite extraction. The tubes were vortexed (15 s) and centrifuged at 21,000$g$ for 10 min at –5°C. The supernatant (900 $\mu$l) was transferred into fresh tubes, recentrifuged at 21,000$g$ for 10 min at –5°C, and the supernatant was removed and dried using a SpeedVac. The samples were stored at –80°C briefly, before analysis by targeted LC/MS/MS to assess the specified metabolite amounts, using methods described earlier (Walvekar et al, 2018a). For methionine uptake experiments, Tregs ($2 \times 10^6$, activated for 38–40 h) were harvested and washed three times with PBS and further cultured in DOSAA and unlabeled methionine (150 $\mu$M) in the absence of IL-2 for 1, 3, and 6 h. 20 min before harvesting and metabolite extraction, the cells were harvested, washed once with DOSAA, and cultured with $^{13}C_5{}^{15}N$-labeled methionine (150 $\mu$M) without IL-2. Subsequently, the relative amounts of labeled methionine (see Q1/Q3 details) and label incorporation from methionine to SAM or SAH were measured using targeted LC/MS/MS based approaches. MS-Q1/Q3 (Parent/Product) parameters used: methionine (Q1/Q3 150.1/56), SAM (Q1/Q3 399/250), SAH (Q1/Q3 385/136), $^{13}C_5{}^{15}N$ Methionine ($5^{13}C$, $1^{15}N$ Q1/Q3 155.1/60), $^{13}C_5{}^{15}N$ SAM ($5^{13}C$ Q1/Q3 405/250), and $^{13}C_5{}^{15}N$ SAH ($4^{13}C$ Q1/Q3 390/136). Mass spectrometer used: AB Sciex 6500 QTRAP.

#### Data normalization

Relative metabolite amounts are typically shown. For this, the first time point (T0) data is normalized to 1, and subsequent samples were compared relative to T0. Raw and normalized data are shown in worksheet 1.

### Real-time PCR analysis

Tregs ($4 \times 10^6$, activated for 38–40 h) were harvested and cultured without IL-2 in the presence and absence of GSI-X (10 $\mu$M) for 3 h. After the incubation, cells were lysed in 700 $\mu$l of TRIzol (Thermo Fisher Scientific). RNA was isolated according to the manufacturer's

instructions. cDNA was synthesized with 1 μg RNA using the Pri-meScript first strand cDNA Synthesis Kit (Takara Bio). cDNA was diluted in a 1:5 ratio and used for real-time PCR using the Maxima SYBR Green qPCR Master Mix and Bio-Rad CFX96 Touch Real-Time PCR Detection System. Relative change in transcript levels was calculated using the $2^{-\Delta\Delta Ct}$ method (Livak & Schmittgen, 2001). Hypoxanthine-guanine phosphoribosyltransferase (HPRT) or actin was used as the reference gene. All primers used are listed in Table S1.

### Apoptosis assay

Tregs were washed with PBS thrice and cultured at $0.3 \times 10^6$ cells/ml 48-well plates in the absence of IL-2 and with the required treatment for 18–22 h. At the end of the treatment, cells were harvested and stained with Hoechst 33342 (1 μg/ml in PBS) in the dark at ambient temperature for 5 min. Cells were washed to remove excess dye, and pellets resuspended in 20 μl PBS and imaged to score nuclear morphology using a fluorescent microscope (Olympus BX-60) (Perumalsamy et al, 2012). Samples were counted double blind, and ~200 cells across several fields were scored for nuclei with normal or apoptotic morphology. Field views showing live and dead cells were acquired using inverted Olympus IX-73, 60× Objective, NA 0.70. Images were processed using Fijji-Image J.

### Immunofluorescence analysis

$0.3 \times 10^6$ Tregs were adhered onto cut confocal dishes coated with 1 mg/ml poly-D-lysine in PBS for 10 min and fixed with 2% freshly reconstituted paraformaldehyde for 20 min in the dark at room temperature. After fixation, Tregs were permeabilized using 0.2% NP-40 in PBS for 10 min at room temperature and blocked with 5% BSA in PBS for 1 h at room temperature. Cells were incubated with a Foxp3 antibody tagged with FITC (1:100) or isotype control antibody diluted in 5% BSA in PBS overnight at 4°C. Cells were washed twice with PBS and stained with Hoechst 33342 (1 μg/ml) for 10 min and 10–15 random fields were imaged using Olympus FV3000 using Plan-Apochromat 63X NA 1.35 oil-immersion objective. Images were processed to remove background based on Isotype controls.

### Statistical analysis and data presentation

Data are represented as mean ± SD of two or three independent experiments (as indicated). Statistical significance was calculated using the two-tailed $t$ test, and $P$-values ≤ 0.05 were considered as significance. Western blots were analyzed using ImageJ software and processed with Adobe Photoshop. Graphs and heat maps were prepared using GraphPad Prism, and figures were prepared using Adobe Illustrator.

## Data Availability

The authors confirm that the data supporting the findings of this study are available within the article (and/or) its supplementary materials.

## Supplementary Information

## Acknowledgements

We acknowledge Freddy Radtke, EPFL, Switzerland, for the Notch1[lox/lox] and Cd4-Cre::Notch1[lox/lox] mice; Barbara A Osborne, Amherst, USA, for pBABE and pBABE-NIC-NES plasmids; and I Verma (plasmid #12371; Addgene) for the pCL-Eco construct. We acknowledge Bangalore Life Science Cluster (BLiSC), the Mass Spectrometry Facility, and Animal Care and Resource Center (ACRC) of NCBS-TIFR and DBT-inStem, Bangalore, India. We especially thank Sreesa Sreedharan for help in completing some metabolite analysis experiments. A Sarin and S Laxman acknowledge support from the Department of Biotechnology (DBT), Government of India grant BT/PR13446/COE/34/30/2015.

### Author Contributions

N Saini: data curation, formal analysis, validation, investigation, visualization, methodology, and writing—original draft, review, and editing.
A Naaz: data curation, formal analysis, validation, investigation, visualization, methodology, and writing—original draft, review, and editing.
SP Metur: data curation, formal analysis, validation, investigation, and methodology.
P Gahlot: data curation, formal analysis, validation, investigation, and methodology.
A Walvekar: data curation, formal analysis, validation, investigation, and methodology.
A Dutta: data curation, formal analysis, and investigation.
U Davathamizhan: data curation, formal analysis, and investigation.
A Sarin: conceptualization, resources, supervision, funding acquisition, visualization, methodology, project administration, and writing—original draft, review, and editing.
S Laxman: conceptualization, resources, supervision, funding acquisition, visualization, methodology, project administration, and writing—original draft, review, and editing.

### Conflict of Interest Statement

The authors declare that they have no conflict of interest.

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
