## [Reviewer comments · Life Science Alliance]

Life Science Alliance

Methionine uptake via SLC43A2 transporter is essential for regulatory T cell survival

Neetu Saini, Afsaana Naaz, Shree Metur, Pinki Gahlot, Adhish Walvekar, Anupam Dutta, Umamaheswari Davathamizhan, Apurva Sarin, and Sunil Laxman

DOI: <https://doi.org/10.26508/lsa.202201663>

Corresponding author(s): Sunil Laxman, Institute for Stem Cell Science and Regenerative Medicine and Apurva Sarin, Institute of Stem Cell Science and Regenerative Medicine

Review Timeline:

Submission Date:	2022-08-09
Editorial Decision:	2022-08-25
Revision Received:	2022-08-30
Accepted:	2022-08-30

Transaction Report:

Please note that the manuscript was reviewed at Review Commons and these reports were taken into account in the decision-making process at Life Science Alliance.

Thank you for the reviews for our manuscript titled “Methionine uptake via SLC43A2 transporter is essential for regulatory T cell survival”. We thank the reviewers for their detailed comments on this manuscript.

We have now addressed essentially all the comments made by the three reviewers. Below is a detailed response letter which addresses these comments, and includes the changes now present in the revised manuscript. For convenience, we have additionally included a ‘marked’ manuscript file with changes in blue.

Additional note: This manuscript should be viewed as a brief/short report. The study makes a novel, exciting observation of a unique metabolic requirement (of methionine consumption) for Tregs survival, under cytokine deprivation. We rigorously assess this phenomenon, and find the core mechanism through which this phenomenon is established in these cells. This study *opens up* several possible lines of future investigation, ranging from molecular mechanisms of regulation of the transporter that allows these cells to take up methionine, to how the Notch1 pathway regulates this amino acid uptake, and the various possible down-stream roles of methionine (via its metabolite, S-adenosylmethionine). These directions are outside of the scope of the current manuscript, and therefore we hope that this manuscript will be viewed as exactly that, an exciting short report.

Detailed responses

Reviewer 1:

1. The authors argue that IL-2 withdrawal induces a series of changes (in the availability of amino acids and therefore in the expression of a series of transporters and solute carriers) that are able to sustain Treg cell survival in conditions of nutrients deprivation. However, what is unclear is whether and how all the described phenomena might change in conditions of optimal growth, therefore in the presence of IL-2. This comparison is pretty important and crucial in understanding the real impact of IL-2 deprivation. Furthermore, to fully confirm the specificity of the entire system, the authors should then re-add IL-2 exogenously to better understand if the observed phenomenon can be reverted. In the absence of such information, it cannot be excluded that the phenomena are independent of IL-2 withdrawal.

Thanks for the comment.

We do not state in the paper that the IL-2 withdrawal itself controls the availability of amino acids/amino acid transporters, and in fact this is evident from the results presented. IL-2 withdrawal itself does not induce changes in the SLC43A2 protein levels. This data is shown as part of a more comprehensive supplementary figure, S5A. Here we see that in wild-type Tregs, SLC43A2 levels remain constant from T0 – T4 (i.e. 4 hours after IL-2 withdrawal).

To summarize, we find that upon IL-2 withdrawal, the amino acid usage specifically for methionine increases, and this requires the SLC43A2 transporter. The SLC43A2 transporter protein depends on NOTCH1 (and not IL-2 withdrawal), and this data is collectively shown in Figures 4C, 4D, and as part of the same supplementary figure, S5A.

We have added several clarifying statements in the results section, primarily related to Figure 4, to fully address this point. We also add a summarizing line in page 14 (lines 295-297), stating that “*The amount of SLC43A2 transporter protein depends on Notch1, and the loss of Notch1 (and not IL-2) reduces SLC43A2.*”

2. The authors should show how these changes in methionine uptake and consumption are associated not only with a different Treg survival rate, but also with functional changes in these cells. Are they less suppressive? How much does methionine impact on the expression of Foxp3 and other Treg-associated markers? All the reported information should be analyzed also functionally, otherwise they could have a fairly minor relevance in a pathological context.

From our analysis there does not seem to be any change in the identity of the Tregs (based on Foxp3 expression), in these different drop-out conditions. There is no substantive change in the fraction of Foxp3 positive cells upon sulfur amino acid deprivation (CMDS, DOSAA+/- IL2), and these data are now included as part of Figure 2, as Fig 2D.

At this stage, the various possibilities of methionine regulating adaptive immune responses will have to be an exciting direction of future enquiry. The discoveries in this study now sets a foundation to be able to do so.

3. In Figure 1B, the authors show a reduction in MET and SAM (methionine-derived product) after IL-2 withdrawal. But why are there no differences in the levels of SAH (another transformation product of Methionine)? Are there any compensatory mechanisms that take place during this process? If so, what are they and how can they impact on Treg cell survival?

Thank you for this comment, which we can clarify.

Fig 1B is a 'bulk estimate' of steady-state levels of any metabolite. When such an experiment is done, the only possible read-out is the 'end-point' accumulation of metabolites, and this does not definitively address either uptake/production, or consumption. Usually, if there are large changes/trends, one can make a predictive statement on use of a metabolite. What we observe here (for steady state amounts) is a large decrease in total methionine in Tregs in their regular medium, after IL-2 withdrawal. In addition, we also see some decrease in SAM, and no obvious decrease in steady state SAH amounts, *in this time frame*. However, this experiment cannot fully account for old/existing pools of each metabolite, or rates of change of each.

In order to definitively address this, a separate 'flux' experiment, usually with the pulsing of stable-isotope/labeled precursors, needs to be performed. This is exactly what we have done, and this is now more carefully shown and explained in Fig 1C, D, E and F, as well as Supplementary Fig. S1. Here, two separate data from a labeling/flux experiment are presented, to address this point. In this experiment, after either 1 hour, or 3 hours or 6 hours of IL-2 withdrawal, a pulse of ¹³C labeled methionine was provided for 20 min (in replaced DOSAA medium+¹³C⁵¹⁵Nmethionine), and intracellular amounts of *labeled* metabolites vs unlabeled metabolites (which come from the pre-existing pool) were measured. Here we see very clear, unambiguous trends for– methionine, its metabolic product SAM, and the product of SAM consumption – SAH. Specifically, at different times after IL-2 withdrawal we provided a pulse of ¹³C labeled methionine, and then accessed the fraction of labeled/unlabeled met present in cells, as well as labeled SAM and SAH. First, we found that after providing labeled methionine even for a brief pulse of 20 min (either at 1, 3 and 6 h after IL-2 withdrawal), essentially all the detected methionine was fully labeled. This indicates that (i) there is high uptake of labeled met in Tregs, and that the existing pool of methionine is constantly, rapidly utilized. To make this point now clearer, we have included a detailed labeling/experimental schematic, as well as detailed MS-MS data (see Fig 1C, 1D and a more detailed Supplementary Fig. S1). Second, from this we measured the amount of labeled (vs unlabeled) SAM and SAH, to specifically address relative rates of methionine consumption (with the implied logic that there will be some difference in the rates of methionine utilization and replacement of existing pools, vs SAM and then SAH). Note that the label on these metabolites can only come from the ¹³C labelled met. Here, we clearly observe that there is a *maximum* amount of labeled SAM at the 1 h time point, and this decreases steadily over the 6 h. This trend is exactly the same for labeled SAH. This data unambiguously reveals that the existing pool of met (which would be unlabeled) is rapidly consumed. We clearly observe that met is converted to SAM (with a large fraction of the SAM pool now being labeled), and that this fraction of SAM is highest in the earliest time window (1 h), and reduces steadily over ~6 h (shown in Fig 1F). In this same experiment, we also see this same trend for SAH, with highest conversion to SAH in 1 h, and this reduces in 6 h (see Fig 1F). This flux experiment therefore unambiguously shows conversion of met → SAM and SAM → SAH, and also indicates that the utilization of methionine is particularly high in this ~1 hour window after IL-2 withdrawal.

We have now included clearer explanations in the text for this experiment, and how this experiment was performed. These are included in the results/conclusions section of the relevant parts of the manuscript (related to Fig 1), on pages 7-9.

4. The authors state that methionine regulates Treg cell survival. The apoptotic rate of Treg cells should always be confirmed by different methodologies (besides microscopic analyzes), such as FACS analysis (for AV and PI positivity). Furthermore, in this regard, the authors should further indicate and analyze which apoptotic mechanisms are put in place, following the IL-2 withdrawal, on which Methionine might directly act. Which molecules are involved in this process? Caspase? death receptors? Analyzing apoptosis in these terms is too general and would benefit from more molecular details.

A detailed characterization of apoptosis in Tregs upon IL-2 withdrawal is reported in our previous studies (*Perumalsamy et al, 2012; Marcel & Sarin 2016*). These studies include microscopic analysis, annexin staining and mitochondrial potential loss, and these studies also have addressed mechanistic details of cell death after IL-2 withdrawal. In this study, we continue to use exactly the same, established experimental paradigm, and assessment approaches. This we now clarify in the revised text in page 9. In addition, images showing apoptotic nuclei (and details on how apoptosis was assessed) have been included as Supplementary Fig. S2.

In this study, we only intend to show that methionine sufficiency prevents cell death upon IL-2 withdrawal, and this requires SLC43A2. We make no further comments about the specific nature of cell death/apoptosis that methionine withdrawal causes, and we feel that this is well beyond the scope of this manuscript, and requires a separate direction of investigation. We acknowledge this in both the results (page 9) and discussion (page 17).

5. The link between methionine and Notch1 does not seem direct and intuitive. The authors should demonstrate more directly what links methionine to Notch1. In this regard, it would be useful and interesting to understand what happens if Notch1 is over-expressed in Notch-deficient mice (Cre+), to understand if the observed phenotype can be reverted. Furthermore, it should be evaluated the frequency and phenotype of Treg cells from Notch1-deficient mice are and how they function, to understand the impact of Notch1 on Treg cell homeostasis and biology.

Thanks for this comment. We have in fact addressed components of these queries, and clarify these in the revision. In Fig 4C and 4D, we include data that indicates that Notch1 deficient Tregs have reduced levels of SLC43A2. Upon over-expression of NIC1-NES (where Notch1 is now enriched in the cytoplasm, see *Perumalsamy Sci Sig 2012*) in these Notch1-null Tregs, the levels of SLC43A2 are rescued, as seen in Fig 4E.

As already reported in earlier studies, the loss of Notch1 (Notch1-null) does not alter the expression or frequency of Tregs. However, Notch1 deficient Tregs show defects in suppressor function (see *Marcel elife 2016*). We have included a clarifying statement in the results section on page 14, to consolidate these disparate facts.

6. Finally, the authors should demonstrate how Notch1 is able to regulate SLC43A2 expression. Is it a mechanism of transcriptional control? How does this regulation happen and work? This aspect is missing and should be better investigated.

Thanks for this comment. At this stage, what we unambiguously find are that SLC43A2 mRNA and protein levels are Notch1 dependent (Fig 3D and Fig 4C and 4D, as well as Supplementary Figure S5A). These data show that in Notch1 null, there is constitutively lower SLC43A2 protein compared to wild-type/Notch1 replete Tregs. Second, this Notch1 dependency appears to be non-canonical, since the expression of NIC-NES (which is cytoplasmically enriched Notch1) in Notch1 null Tregs, restores the expression of SLC43A2 (Fig 4E).

As an additional control, we have also analyzed the SLC43A2 promoter for putative RBP-jk binding sites. However, the promoter of SLC43A2 does not have any RBP-jk binding site, and also therefore hints at non-canonical Notch1 function regulating SLC43A2. This is now included as a statement in the manuscript on page 15. This prompted us to further very briefly explore this possibility, and we include further data with the NIC-NES (cytoplasmically enriched Notch1) that shows rescue of SLC43A2 protein in Notch1 null Tregs (consistent with this possibility). Collectively, these data suggest non-canonical Notch1 dependence of SLC43A2.

The study of the detailed mechanisms of non-canonical Notch1 mediated control of SLC43A2 will be an exciting area of future research, and is likely to open up many new mechanisms, but is beyond the scope of this current study.

Reviewer #2 (Evidence, reproducibility and clarity (Required)):

The study used multiple methods (inhibitor, ShRNA, gene knockout) to elicit the mechanism by which Tregs upregulate the expression of SLC42A2 through Notch 1 signalling under IL-2 deprivation, which is required for Treg survival in the absence of IL-2. Overall the results are convincing.

Major

However, a few experiments need to be conducted to strengthen the study.

The authors stated that the decreased methionine level in Treg under IL-2 deprivation "suggested a shift in these cells, towards increased methionine uptake and consumption." However, This is not necessarily true, the decreased methionine level can also be a result of decreased methionine uptake. To prove the statement, Isotopic labelling experiments should be conducted to directly measure the methionine uptake and metabolism in Tregs before (i.e. T0) and during IL-2 deprivation (i.e. 1h, 3h, and 6h of treatment). However, the former was not studied.

Thank you for this comment.

We have in fact carried out systematic pulse-label based experiments with ¹³C labeled methionine. We apologize that this section, and the conclusions that could be drawn were (in retrospect) insufficiently explained in the original submission. As also explained earlier (to a point raised by reviewer 1), we have now substantially clarified this point, and also include detailed explanations (and data) for these experiments, to establish that Tregs take up and utilize more methionine (in a temporal manner), post IL-2 withdrawal.

To briefly summarize the response included earlier: a separate 'flux' experiment, with labeled precursors (¹³C-methionine), was performed, and the results of this experiment are now more clearly shown in Fig 1C, 1D, Fig 1E and 1F, as well as Supplementary Fig. S1. Two separate sets of data are presented, to address this point. Here we see very clear trends for all metabolites – methionine, its metabolic product SAM, and the product of SAM consumption – SAH. Specifically, at different times after IL-2 withdrawal we provided a pulse of ¹³C labeled methionine, and then accessed the fraction of labeled/unlabeled met present in cells, as well as labeled SAM and SAH. First, we found that providing labeled met for a brief pulse of 20 min, at 1, 3 and 6 h after IL-2 withdrawal, replaced all the unlabeled met in this short pulse time. This indicates that there is high uptake of (labeled) external met, and (ii) that the existing pool of met (which would be otherwise unlabeled) is rapidly consumed/replaced. To make this point now clearer, we have included a detailed labeling/experimental schematic, as well as MS-MS data (Fig 1D, Fig 1E and Supplementary Fig. S1). Second, we follow this labeled methionine, as it is converted to (labeled) SAM and SAH (where the label can only come from the ¹³C labeled met). This experiment was done to specifically assess consumption of methionine. Here, we clearly see that (i) met is converted to SAM (with a large fraction of SAM now being labeled), and that this fraction of SAM is highest in the earliest time window (1 h), and reduces at ~6 h (Fig 1F). In this same experiment, we also see this same trend for SAH, with highest conversion to SAH in 1 h, and this reduces in 6 h (Fig 1F). This flux experiment therefore unambiguously shows conversion of met → SAM and SAM → SAH, and that this met consumption is highest at 1 h post IL-2 withdrawal.

We have now included clearer explanations for this experiment, and included these results/conclusions in the relevant results section of the manuscript (related to Fig 1).

What is the mechanism by which Notch 1 induces apoptosis in Tregs? Can the apoptotic death be rescued by a pan-apoptotic inhibitor such as z-VAD-FMK?

Thank you for this comment, which overlaps with a comment made by reviewer 1. A detailed characterization of apoptosis in Tregs upon IL-2 withdrawal was carried out extensively in our previous studies (Perumalsamy Sci Sig 2012, Marcel eLife 2016 etc). These studies had included microscopic analysis, annexin staining and mitochondrial potential loss, and have addressed details of cell death after IL-2 withdrawal. In the present study, we use exactly the same experimental paradigm, with no changes whatsoever, and also assess cell death microscopically. This we now clarify in the

revised text in page 9. In addition, images showing apoptotic nuclei (and details on how apoptosis was assessed) have been included Supplementary Fig. S2. At this stage a further analysis of apoptosis in these cells is beyond the scope of this manuscript, but we acknowledge this in both the results (page 9) and discussion (page 17).

To what extent is the death by IL-2 withdrawal due to other metabolic disruptions? For example, do glucose levels change as a result of reduction in GLUT1? In other words, how specific is the effects due to methionine because the methionine rescue is 100%. At a minimum this should be carefully considered and the main conclusion be potentially tempered if such a possibility exists. ##### At this point we make no claims on other metabolic disruptions. What we unambiguously show is this increased use of methionine after IL-2 withdrawal, and the requirement of this amino acid specific transporter for Tregs survival. Of course, there can undoubtedly be other metabolic changes (including particularly glucose utilization or metabolism) in Tregs upon IL-2 withdrawal. Methionine, primarily through its metabolic product SAM, controls a variety of fundamental metabolic, signaling and regulatory processes in various cells. Diverse recent studies suggest that the presence/absence of methionine triggers cascading, hierarchically organized metabolic programs (including changes in glucose metabolism), to alter proliferation vs survival trajectories, which would be exciting future possibilities. In fact, a highlight of our study is showing this unique methionine-dependent metabolic program in Tregs, which has not been studied before, and clearly suggests that this is very different from what is already known in other T-cell populations such as Teffs. Separately, growing evidence in other systems (including some of our own research) points to how important a role methionine metabolism plays in overall carbon flux, including glucose/pentose phosphate metabolism, and so it is entirely plausible that there are cross talks and cascading effects. Indeed, we really hope that this study will actually stimulate and expand other studies of metabolic control of cell states in Tregs, to address these possibilities. We allude to these in an expanded discussion on page 17 and 18.

In the Notch KO (Fig 3) and transporter KD experiments (Fig 4), the authors should show that quantitatively methionine uptake and not other amino acid is blocked. In both cases, does supplementation with any methionine cycle metabolite (e.g. SAH, SAM, etc) rescue from apoptosis? Does overexpression of SLC3A2 result in more methionine under IL-2 withdrawal?

Thanks for this comment. We have indeed considered this possibility. However, we have collective evidences from other experiments to minimize this possibility (or suggest that it is peripheral to the phenomenon being described here). First, in Fig 2A, we unambiguously show that adding methionine alone uniquely rescues Tregs cells from death. Furthermore, in Fig 4B, we find that the knock-down of SLC43A2 Tregs *cannot* be rescued from apoptosis by the supplementation of methionine (while control/scrambled treated Tregs were rescued). This shows that (i) SLC43A2 is required for Tregs survival upon IL-2 withdrawal, and (ii) for the methionine-mediated rescue. Since the SLC43 class of transporters are well known to transport methionine, we have to make a collective inference as to this rescue.

The authors showed that GSI-X abolished the upregulation of SLC42A2 induced by IL-2 deprivation (Fig 3D). However, GSI (7.5uM) seems to be very toxic to Tregs, it caused nearly 40% apoptosis in Tregs (30% more than control) (figure 3A). Therefore, it is unclear whether the inability of Tregs to upregulate SLC42A2 expression under "No IL-2 + GSI" condition was caused by GSI cytotoxicity or by Notch 1 inhibition (figure 3D). I would suggest the authors to compare the expression of methionine transporters in Notch 1-knockout and Notch 1 competent Tregs, which have a similar viability as wildtype Tregs.

Thanks for this comment. It is important to note that the estimates of death after treatment with GSI (in Fig 3A) is from a duration of ~18 h. However, for all the analysis of the SLC transporter mRNA (Fig 3D), these are done within ~3 h of GSI treatment. Therefore, the fraction of death in this short time cannot be related to the data shown in Fig 3A. The only point of Fig 3A is to show that Notch1 signaling is required for the methionine mediated rescue of Tregs survival.

Separately, in Fig 4C and 4D, and Fig. S5A, we assess the protein levels of SLC43A2 in Notch1^{+/+} (wild-type) and in Notch1-null Tregs. Here, we find that SLC43A2 (and not SLC7A5, Fig. 4C) protein levels are constitutively lower in the Notch1-null cells. Collectively, these data strongly suggest that SLC43A2 depends on Notch1 signaling.

Minor:

Fig 1Dii - it appears that SAM and SAH levels decrease almost equally between 1h-3h and 3h-6h. The highest uptake and utilization over the first 1 h after IL-2 withdrawal as the authors state is not apparent. Also the error bar for SAH is large yet the statistical analysis shows $p < 0.05$? Can the authors please confirm these two points?

We have made these clarifications in the revised text around Figure 1. Also see earlier responses.

The authors did not clarify whether and how they assessed the purity of in vitro-maintained Tregs before the cells were used for assays. This is necessary because beads-sorted cells are usually not 100% pure and the CD3/CD28/IL-2 activation condition used in the study can also support the growth of other T cell subsets.

This protocol for Tregs isolation has been extensively optimized in earlier studies (*Marcel & Sarin 2016, Saini et al, 2022*). In this revision, we have also now included Foxp3 staining in Tregs subject to these various condition, as shown in Fig 2D and Supplementary Fig S4E.

Reviewer #2 (Significance (Required)):

New study that demonstrates the involvement of Notch1 in amino acid regulation of metabolism and survival of T cells.

Reviewer #3 (Evidence, reproducibility and clarity (Required)):

For transparency, we have already received an earlier version of this paper for review.

As the purpose of Review comms is to avoid re-reviewing papers in this way, we are happy to reproduce that review here, amended as appropriate with the evolution of the manuscript.

Saini et al. aim to dissect the metabolic requirements of regulatory T cells for their survival. They identify the uptake of methionine as essential for the survival of antigen activated regulatory T cells (Tregs) following IL-2 deprivation. They conclude that this process is regulated by Notch1 activity which in turn controls the expression of the SLC43A2 protein, a methionine transporter, upon IL-2 withdrawal.

The data and the methods used are for most part clearly presented and described. However, there are some points that need to be resolved for clarity.

Figure 1A inset schematic, and materials and methods - it is not clear what conditions were used to activate/differentiate the Tregs.

"Cells were activated for 38 h with 20 ul/ml magnetic beads coated with antibodies to CD3 and CD28" Is IL-2 present during the full activation time and in what amount?

No IL-2 was present during activation. Clarifications included now in the text in the methods section (page 19).

Anything else added to the culture? E.g TGFb

nothing else is added to the culture. Now cleared in the methods section (page 19).

In the M&M session describing the retroviral transduction assay cells were cultured in 1 ug/ml of IL-2. Is this the concentration always used? What is the rationale behind this choice?

Yes. We have used 1ug/ml concentration of IL-2 after retroviral transduction to minimize stress. This statement is included in the methods (page 22).

Figure 1D (i) and (ii)

"Met % labelled" would be clearer if clarified that is relative to T0

This is now clarified more extensive in the figures and the text. The % label is not relative to T0, but the fraction labeled/unlabeled at that time point, since this is a flux type experiment. At T0, there is no detectable label for methionine, since there is only unlabeled methionine in the cells. A more detailed supplemental fig S1 now shows the actual LC/MS/MS peaks (and overlays), to illustrate how these analysis were done, as a stable-isotope pulse experiment. The y-axis is more clearly labeled now, and the data more collectively presented as Fig 1E and Fig 1F. In Fig 1E, the y-axis is % labeled (for methionine), which comes from the relative amount of labeled methionine in these cells, vs unlabeled.

In Fig 1F, we are assessing how much label from met ends up in SAM or SAH (to indicate the consumption of met), so the % label is that of labeled/unlabeled SAM or SAH (as now indicated).

Figure 2

There needs to be more clarity about the cells used in the experiments

i) Were Treg cells used in these experiments activated for 38 h and then cultured for 18-22 h in the absence of IL-2?

Yes, this is correct.

The data show just above 10% of apoptotic nuclei were scored when Tregs were cultured in complete media without IL-2. The figure should also include the % of apoptotic nuclei of Tregs cultured in complete media in the presence of IL-2.

We have earlier established that WT Tregs (sufficient in Notch1) do not show any significant difference in apoptosis in the presence or absence of IL-2 (*Marcel and Sarin 2016, Marcel et al 2017*).

In Fig 2 we have only tried to establish the requirement of methionine in the survival of WT Tregs in the absence of IL-2.

Notch1 nulls on the other hand do not survive in complete media without IL-2. That is why in Fig 3B, where we compare the requirement of methionine for cell survival in Notch1 nulls vs Notch1 replete Tregs we have included the % of apoptotic nuclei of Notch1 null Tregs cultured in complete media in both presence and absence of IL-2.

Do note that in Fig 2, the data shown are from WT Tregs, and in Fig 3B, the data show the difference in apoptosis in the presence or absence of IL-2 in Notch1 null cells.

Figure 3A

The amount of methionine used in panel A is higher (200 μ M) than the amount used in the rest of experiments presented in this manuscript (150 μ M). Can the authors explain this rationale for this and discuss if appropriate.

In this experiment (Fig 3A), the addition of methionine was not done at the onset of IL-2 withdrawal as in earlier experiments. Therefore, just to ensure methionine-dependent protection of Tregs, we added a slightly higher concentration (200 μ M) of methionine. We have clarified this on page 11 in the revision.

Figure 3B

The graph shows the % of apoptotic nuclei in Tregs Notch 1 lox/lox CD4 Cre positive versus negative. The same colour scheme is used in both cell genotypes however the conditions linked to each colour are different. This makes the figure confusing.

It would aid clarity of the figures colour scheme were modified.

We have now added a modified, consistent color scheme to all the bar graphs, in order to make interpretations unambiguous. We have also now depicted protein/western blot quantifications in black/grey, to prevent any confusion.

In both conditions, are the cells IL-2 deprived?? The IL2 presence/absence needs to be clearly stated

now clarified in the figure

Materials and Methods

The length of activation time changes between experiments (from 38 up to 44 h) so does the volume of magnetic beads used (15-20 μ l/ml - Is this meant to be μ g/ml?).

Can the authors explain the reason of this choice?

this was in an earlier version of the manuscript. As the number of Tregs activated were reduced, there was a proportional reduction in the activation beads used. This is no longer applicable to the current version of the manuscript.

Statistical analysis

"Data are represented as Mean {plus minus} SD of two or three independent experiments".

Does this mean two or three samples in total or three samples in each experiment?

This needs to be made clearer in the Figure legends

Done

August 25, 2022

RE: Life Science Alliance Manuscript #LSA-2022-01663

Dr. Sunil Laxman
Institute for Stem Cell Science and Regenerative Medicine (inStem)
Regulation of Cell Fate
GKVK post
Bangalore, Kar 560065
India

Dear Dr. Laxman,

Thank you for submitting your revised manuscript entitled "Methionine uptake via SLC43A2 transporter is essential for regulatory T cell survival". We would be happy to publish your paper in Life Science Alliance pending final revisions necessary to meet our formatting guidelines.

- please upload your supplementary figures as single files and upload your table files in editable doc or excel file format
- please add a Running Title, Summary Blurb, and category to our system
- please add the Twitter handle of your host institute/organization as well as your own or/and one of the authors in our system
- please add the author contributions to the main manuscript text
- please use the [10 author names, et al.] format in your references (i.e. limit the author names to the first 10)
- please add your supplementary figure legends and table legends to the main manuscript text

Figure Check:

- please provide the original blots for Figure S5 as Source Data

A. FINAL FILES:

B. MANUSCRIPT ORGANIZATION AND FORMATTING:

Sincerely,

Reviewer #2 (Comments to the Authors (Required)):

Overall the authors' revisions have satisfied my concerns. I appreciate the new experiment work especially the methionine tracing studies. While these experiments do not fully address mechanism, they are confirmatory to the observed phenotype in the T regs.

The paper provides new insights and advances that will be of interest to the broad community.

August 30, 2022

RE: Life Science Alliance Manuscript #LSA-2022-01663R

Dr. Sunil Laxman
Institute for Stem Cell Science and Regenerative Medicine
Regulation of Cell Fate
GKVK post
Bangalore, Kar 560065
India

Dear Dr. Laxman,

Thank you for submitting your Research Article entitled "Methionine uptake via SLC43A2 transporter is essential for regulatory T cell survival". It is a pleasure to let you know that your manuscript is now accepted for publication in Life Science Alliance. Congratulations on this interesting work.

DISTRIBUTION OF MATERIALS:

Again, congratulations on a very nice paper. I hope you found the review process to be constructive and are pleased with how the manuscript was handled editorially. We look forward to future exciting submissions from your lab.

Sincerely,
